# Anticipating Changes in Lifestyles That Shape Travel Behavior in an Autonomous Vehicle Era—A Method-Oriented Systematic Literature Review

**Thomas Le Gallic [1] and Anne Aguilera [2,*]**

[1] CIRED, CNRS, 94130 Nogent-sur-Marne, France; thomas.le-gallic@enpc.fr
[2] Laboratoire Ville Mobilité Transport (LVMT), Université Gustave Eiffel and ENPC, 77454 Marne-la-Vallée, France
\* Correspondence: anne.aguilera@univ-eiffel.fr

**Abstract:** This article proposes a systematic review of the recent literature on the impacts of the deployment of autonomous vehicles through the lens of lifestyle changes that will modify our mobility practices. It discusses the main findings of the studies, analyzes their links with the foresight methods used by their authors, and identifies research gaps. Four components of lifestyles are considered: residential location, car ownership, activity patterns, and tourism. Particular attention is given to the diversity of the reviewed foresight approaches, to the way they complement one another in the construction of knowledge, and to their influence on the forecasts and the lessons learned. Our work shows a convergence of results across methods, especially for expected impacts on household location and car ownership, and the influence of the way autonomous vehicles are deployed. Our analysis also makes it possible to draw up a more comprehensive and nuanced picture of the anticipated changes in lifestyles, and the main sources of uncertainty. Finally, our work identifies several research gaps and avenues for future studies such as the impacts on job choices and tourism, the need for a better understanding of the links between the different deployment models for autonomous vehicles, and the need to explore more scenarios that are compatible with environmentally- and socially-oriented goals.

**Keywords:** autonomous vehicles; lifestyle changes; mobility; foresight methods; literature review

## 1. Introduction

Autonomous vehicles (AVs) are heralded as a transformative technology which will drive significant social change. By reframing the terms of accessibility—through changes in the time we are willing to spend on travel, travel speed, and transportation costs—autonomous vehicles are likely to change our lifestyles in many ways with important consequences on the various aspects of our daily and long-distance mobility practices (especially frequency, average distances, and modal choice).

Anticipating these changes, i.e., foreseeing the future and preparing for it [1], is of critical importance in order to regulate the deployment of autonomous vehicles and thereby mitigate their negative environmental and social impacts and foster their benefits [2,3]. The medium- and long-term decisions at stake relate to innovation policies, but also to planning in terms of infrastructure investments and land-use policies. Foreseeing these changes nonetheless represents a considerable scientific challenge since, on the one hand, the technology is not mature and, on the other, lifestyles are the result of complex, interdependent decisions involving many factors and a variety of timescales.

After a period when the literature was mainly focused on technological aspects [4], studies exploring the social and societal implications of autonomous vehicles have proliferated in recent years, as evidenced, for example, by the bibliometric review by [5]. Various issues have been considered, such as acceptance [6,7], mobility practices e.g., [8], vehicle

ownership [9], the nature and spatiotemporal organization of activities e.g., [10], residential strategies [11], or impacts on health and well-being [12]. Some approaches have considered the more radical changes that could affect the transport system in the future, e.g., [13] and the shape of cities [14,15], as well as the political and societal implications of autonomous vehicles [16]. Finally, some authors have raised new modelling issues [17] and, in particular, that of modelling the interactions between transport and land use [18].

In this paper, we provide a new perspective on social changes brought about by the deployment of autonomous vehicles based on a systematic review of studies that foresee changes in lifestyles. This is the first time such a lifestyle framework has been adopted, and by doing so, this study addresses a set of interdependent life choices that shape mobility practices. The purposes are to discuss the main findings, analyze how these are linked to the foresight methods used by the authors, and identify research avenues. Beyond the novelty of this framework, this study highlights the value of a multi-method review. To this end, this paper is marked out by the attention given to the diversity of the reviewed foresight approaches, to the way they complement one another in the construction of knowledge, and to their influence on the forecasts and the lessons learned. It is important to note that this review concentrates on studies that assume that highly autonomous vehicles will become widespread on most road networks, which is the only situation that seems likely to lead to significant changes in life choices [19].

The remainder of the article is organized as follows. Section 2 introduces the conceptual framework of lifestyles, describes the constitution of our corpus, and explains how we classified the various methods used by the authors. The corpus is analyzed in Sections 3 and 4. Section 3 describes the changes anticipated in the autonomous vehicle era for each of the components of our lifestyle framework, and analyses how these are linked to the method used. In Section 4, we discuss the main insights from this multi-method review and provide new perspectives for future research.

## 2. Materials and Methods

Our general approach follows the methodological principles for literature review papers and systematic reviews recommended by [20,21], respectively. It includes, in particular, (a) planning activities; (b) realization; and (c) reporting activities, including a discussion of our findings. This section will describe the framework that shaped our analysis, our corpus and the way we classified the foresight methods we encountered.

### 2.1. Lifestyle Framework

Lifestyle is a commonly used notion in disciplines such as sociology, psychology, economics, human geography, and marketing to describe and analyze how human beings organize their lives. It encompasses multiple dimensions, from eating habits to domestic arrangements, and from leisure activities to attitudes towards work or consumer practices. In this article, we propose a framework that consists of five interdependent components which both influence and are influenced by travel behaviors. These components concern: (a) choices about housing and especially residential location, (b) choices about car ownership, (c) choices about jobs and incomes, (d) choices about day-to-day activities, and (e) choices relating to vacations. A full description of each of these components is given in Appendix A.

Figure 1 provides a representation that highlights key interdependencies between these components and with travel behaviors. Firstly, these components are linked by three "affordances" accessible to an individual: free time, financial resources, and mobility, i.e., the capacity to be mobile [22]. The first three components involve particularly influential choices, generally covering a timeframe of a year or several years, that determine the availability of the three affordances. The other two components include choices about how they are allocated to the accomplishment of activities, which are divided into two main categories: day-to-day and more exceptional. All these choices are therefore linked by trade-offs in time and space, whereas mobility practices make the activities spatially possible.

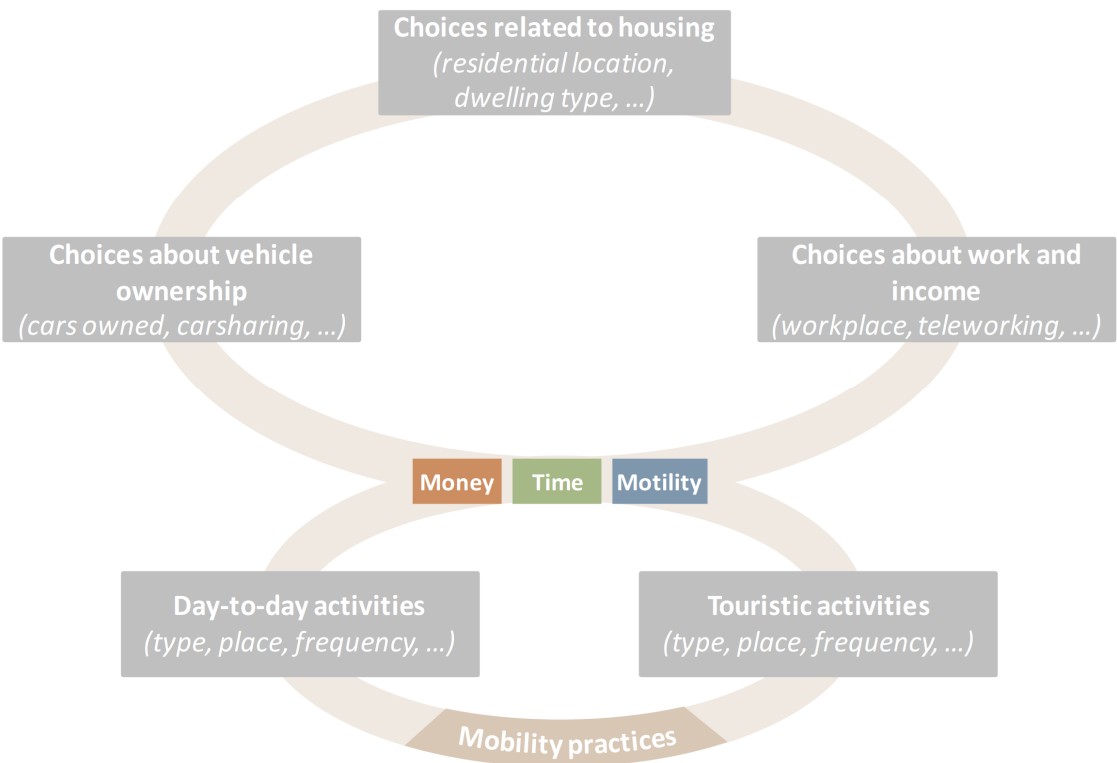

**Figure 1.** Representation of the lifestyle framework: components and interdependencies. Source: Authors.

*2.2. Constitution of the Corpus*

The previous framework was used to build our corpus and also structure our analysis. Our material was assembled in three stages (see Figure 2). Firstly, we identified a set of books, chapters, and papers dealing with autonomous vehicles from searches in the Scopus and Web of Science databases (keywords: "vehicle automation" OR "autonomous vehicle" OR "autonomous car" OR "self-driving vehicle" OR "self-driving car" OR "driverless vehicle" OR "driverless car" OR "automated vehicle" OR "automated car" OR "automated driving" OR "robocar" in selected fields (see Appendix B)). Our most recent update was in March 2021. Secondly, we pre-selected, on the basis of the titles and abstracts, the publications and papers likely to deal with the effects on the five components of lifestyle described below. Due to a lack of studies (see also the Discussion section), it proved impossible to cover the component relating to choices of job and incomes. Thirdly, we analyzed the content of the pre-selected publications to assess whether we would actually include them in the corpus: for us to do this they had to include at least one forecast of the effects of the spread of autonomous vehicles on at least one of the five lifestyle components. In accordance with our lifestyle framework, this definition is precise and fairly restrictive. For example, it involves excluding many studies that aim to provide insight into attitudes towards ownership versus shared mobility, e.g., [23,24], as they address preferences between the two modes, but not explicitly the decision to own (or relinquish) a car. Within this process, and following the same criteria, we consciously added conference papers and reports cited in the reviewed publications but not present in the Scopus and Web of Science databases. This extension sought to enlarge our corpus and, in particular, to diversify the pool of perspectives and methods used. However, for the purposes of transparency, we systematically specified the type of publication in the results section (i.e., paper in a peer reviewed journal, conference paper, chapter, book, or other). In the end, our material consisted of 51 publications (34 journal papers, 2 book chapters, 7 conference papers, 8 reports) representing 49 studies (two studies are described in two publications).

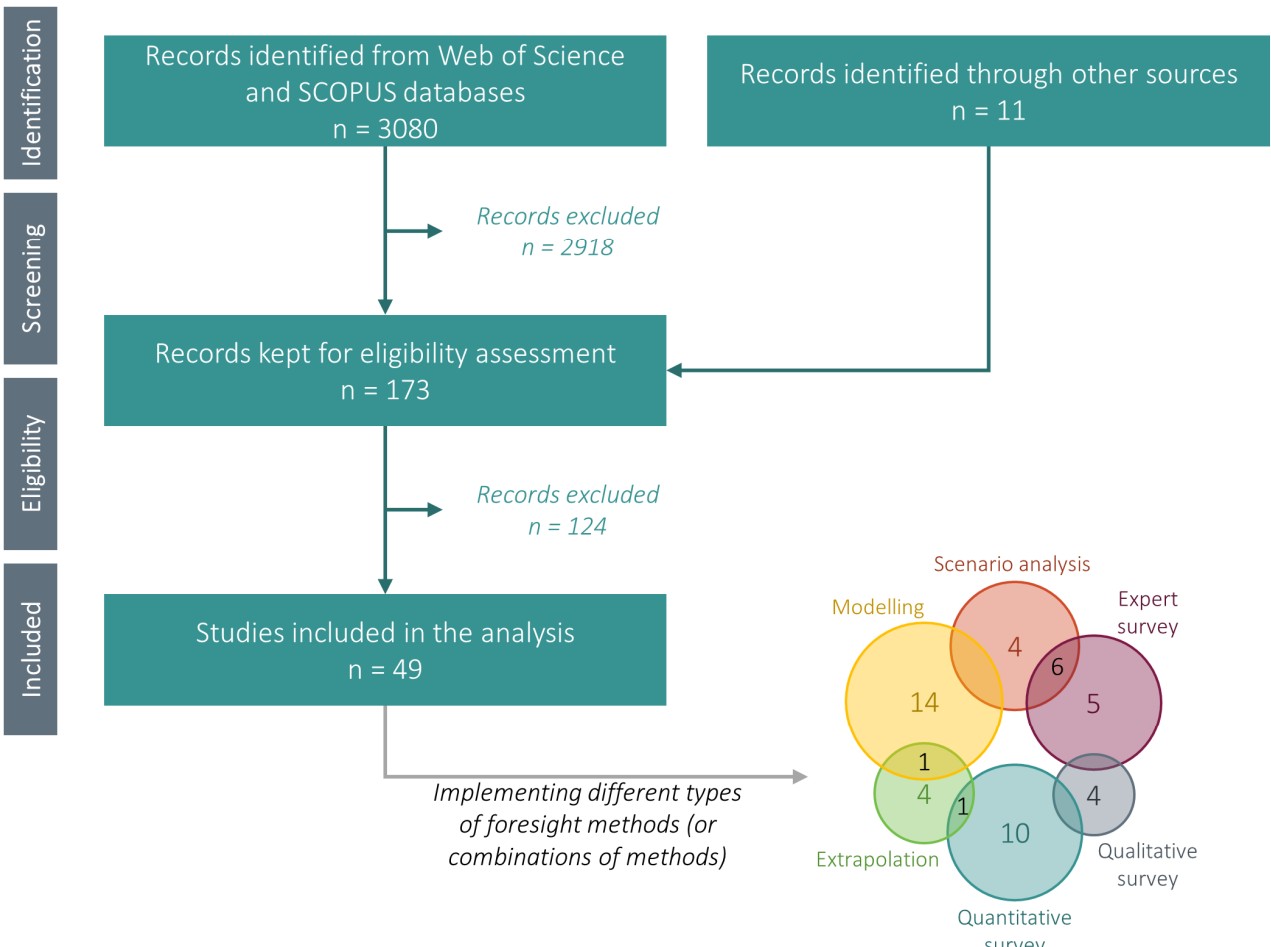

**Figure 2.** Identification and selection process of studies in our corpus, and number of studies according to family of foresight approaches. Source: Authors.

### 2.3. Classification of the Foresight Methods Used

As expected, a wide variety of methods are used in our corpus. In order to capture and discuss their characteristics and contributions, we propose to divide them into six families: (a) modelling approaches; (b) extrapolation approaches; (c) quantitative or (d) qualitative surveys on potential users; (e) expert surveys; and (f) scenario analysis approaches. We classified all the studies in our corpus according to the family (or the two families) they belonged to (see Figure 2). When several methods were used for different purposes, we considered the method(s) specifically used to inform the component of lifestyle at stake. Some methods were considered as hybrids because they combined two families of methods.

The first family (modelling approaches) is based on models which provide a formal representation of a system (which includes, for example, an area with its transportation system or a population with its travel behaviors). Such tools enable the authors to represent and simulate changes and effects, such as changes in accessibility through the notion of generalized transport cost. These approaches make it possible to represent, in particular, a new transport mode, which is represented by a set of characteristics (e.g., travel speed, road capacity, value of travel time (generally to represent changes in convenience or comfort), and direct transport costs).

The second family (extrapolation approaches) consists of estimating changes in behaviors by analyzing current behaviors as reported in existing surveys on current practices and making hypotheses about how accurately they describe the future. This approach has some features in common with modelling approaches but does not rely on formalized (and pre-existing) models. The central hypothesis is usually based on an analogy between an ex-

isting transport practice or mode and the image one has of future autonomous vehicles. For example, public transport is an existing mode in which passengers can perform activities other than driving, as imagined for the autonomous vehicle. Current ride- or car-sharing services can also be seen as having many features in common with future autonomous vehicles (see discussion in [25]).

The third and fourth families (quantitative or qualitative surveys) consist of surveys where individuals are asked about probable changes in their behavior (e.g., residential location, car ownership), for instance, in stated preference surveys. In these, individuals are presented with a fictitious situation, with a view to understanding their representations of autonomous vehicle technology and its consequences on their lives. Approaches that belong to these third and fourth families mainly differ in terms of the number of respondents (from hundreds to tens of thousands in the first case and from tens to hundreds in the second), the ways of interviewing (for example, mainly close-ended questions in the first case and open-ended questions, sometimes as part of a collective process, in the second), and the ways of analyzing the responses (e.g., statistical analysis or textual analysis).

The fifth family (expert surveys) consists of interviewing experts about the effects they anticipate. It encompasses several modalities of questioning and analysis, including collective or individual processes, predominantly textual or predominantly quantitative analyses and a variable number of respondents. It is often combined with the approaches applied in the sixth family.

The sixth family (scenario analysis) consists of building and analyzing scenarios in which autonomous vehicles are widely adopted. This approach is based on at least two principles. Firstly, the use of expertise and knowledge, either by involving experts in the process (in which case developing scenarios becomes a collective process), or by analyzing the existing literature. The experts generally have good knowledge of the system at stake (transport and mobility systems, land use issues), and potentially of the autonomous vehicle. Secondly, the use of cognitive tools to structure thought and stimulate the imagination. Several combinations of methods and processes have been used in the reviewed studies, and most include narratives to reinforce the coherence of the futures that are considered and contrasted scenarios to anticipate a wide range of possibilities. The results can take various forms, for example conceptual models, quantifications, and storylines.

## 3. Results

For each of the four lifestyle components described below, the results are presented by crossing three main types of information: (1) the impacts of autonomous vehicle deployment that are recurrently foreseen by the literature; (2) the type of deployment that leads to these effects (shared or private vehicles); and (3) the method that investigates this effect. We constructed summary tables for each lifestyle component to cross-reference these three types of information. The extended tables on which they are based are available in Supplementary Materials, providing useful details, nuances, and further information on each reviewed case that the synthesis sometimes omits for the purposes of simplicity. As this information is also reported below, it is worth noting that we also analyzed potential similarities and differences across the countries in which were conducted the studies. However, we did not find any significant trend in our corpus, the type of deployment appeared to be much more explanatory. We can only highlight a slight over-representation of the private ownership assumption in US studies.

### 3.1. Residential Location

Residential location choices are long-term decisions, and are part of the life cycle of individuals and households. They are shaped by strong structural constraints, including urban dynamics that land-use and infrastructure policies aim to control. Households have to deal with multidimensional trade-offs, for example rental costs (incomes, dwelling size) and accessibility. These decisions are influenced by the available transport modes and

their features—average speed, convenience, comfort, and costs (out-of-pocket, direct or indirect)—define the terms of accessibility to activities, jobs, and services.

The studies in our corpus concur to some degree with regard to this choice, but not completely (Figure 3). When generalizing them, the results are polarized around two responses that reflect a dominant forecast: dense areas are preferred when autonomous vehicles are shared, and sparsely populated areas are preferred if they are privately owned. The potential effects of the deployment of autonomous vehicles are therefore contrasting, as in approximate terms these effects are opposed (even if densification in the city center and urban sprawl can occur at the same time in different areas). We shall describe the results according to the type of methodological approach in the following.

Qualitative and quantitative surveys on potential future users of autonomous vehicles reveal intentions to relocate, albeit in varying proportions depending on the study. Only a few of the individuals interviewed in the qualitative study conducted by [26] considered that the availability of autonomous vehicles could have an impact on their location choices (see Supplementary Materials for the detailed results). The large quantitative survey conducted by [27] highlights that about 11% of respondents would move further from the location they travel to most often and 12% would move closer (NB: this survey did not raise the issue of whether the autonomous vehicle was privately owned or shared). Considering a private ownership hypothesis, ref. [28] found that 32% of respondents would be ready to move home with a longer commute time. Finally, and contrastingly, the empirical study led by [29] concluded that no long-term changes in the valuation of travel time due to the advent of AVs should be expected to be reflected in residential location choice.

Two other quantifications that are in line with the dominant forecast are provided by studies based on the extrapolation of existing survey data. By extrapolating data from the US National Household Transportation Survey (NHTS), ref. [30] found significant changes in population distribution (a reduction of between 2.8% and 6.1% in rural areas and an increase of between 2.3 and 12.0% in urban areas) when considering a future "where shared, on-demand AVs are mostly suited for short trips in highly urbanized environments" [30]. Furthermore, ref. [31] built estimations of household relocation for a city based on an ad-hoc survey and also found significant trends to relocate to suburban areas.

Although they relied on different formalisms and came from a variety of theoretical perspectives, the modelling approaches gave fairly homogeneous results, forecasting that the deployment of private autonomous vehicles would lead to significant proportions of the population deciding to move away from urban centers [32–38]; whereas they expected the deployment of automated mobility-on-demand systems (AMOD) to lead to an increase in the attractiveness of dense areas [11,32,34,35,37]. In the first case, the key factor was the lower generalized cost of transport or the higher average speed (e.g., when road capacities are increased), which justifies longer daily trips; in the second case, the key factor was access to a more efficient mobility service in dense areas, and in one case, parking land reallocation [34]. Moreover, the study conducted by [39] came to less certain conclusions. The authors concluded there would be no effects on urban sprawl in private ownership scenarios, but a slight positive effect on urban sprawl in another scenario considering both privately owned and shared autonomous vehicles.

The results of most of the scenario analysis approaches (with or without an expert-based survey) also agreed with the dominant findings (Figure 3), albeit with some reservations and nuances. For example, one of the scenarios described by [40] anticipated the attractiveness of low-density areas with a combination of autonomous private cars and automated taxis, but estimated that this dynamic could be largely tempered by land-use policies. In addition, ref. [41] considered that a positive impact on the attractiveness of high-density areas would require a high level of coordination between the on-demand transport system and the public transport system.

| Study | Journal, Conference, Book or Other | Type of method | Increasing population in denser area | Increasing population in suburban area | Study area [when unspecified, countries of the authors to inform the geographical context] |
|---|---|---|---|---|---|
| Zmud and Sener, 2017 | C | Qualitative survey | N | N | Austin, USA |
| Krueger et al., 2019 | J | Quantitative survey | *no change (P)* | | Sydney, Australia |
| Kim et al., 2020a | J | Quantitative survey | N | N | Georgia, USA |
| Moore et al., 2020 | J | | | P | Dallas, USA |
| Carrese et al., 2019 | J | Quant. surv. & extrap. | | N | Roma, Italy |
| Bin-Nun and Binamira, 2020 | J | Extrapolation | S | | USA |
| Kim, 2015 | O | | | P | Seoul, South Korea |
| Thakur et al., 2016 | C | | S | P | Melbourne, Australia |
| Zakharenko, 2016 | J | | | P | Hypothetical city |
| Gelauff et al., 2017, 2019 | O, J | | S | P | The Netherlands |
| Zhang and Guhathakurta, 2018 | J | Modelling | S | | Atlanta, USA |
| Medina-Tapia and Robusté, 2018 | C | | | P | Hypothetical city |
| Larson and Zhao, 2020 | J | | S | P | Average city, USA |
| May et al., 2020 | J | | S | P | Leeds, UK |
| Cordera et al., 2021 | J | | | P , S | Bay of Santander, Spain |
| Townsend, 2014 | O | | S | P | Cities in USA |
| Heinrichs, 2016 | B | Scenario analysis | | P | [Germany] |
| Papa and Ferreira, 2018 | J | | S | P | [UK and Portugal] |
| Gruel and Stanford, 2016, 2017 | C, B | | | P | [USA] |
| Milakis et al., 2017b | J | Scenario analysis supported by experts (workshop, survey) | | P , N | The Netherlands |
| Saujot et al., 2018 | O | | S | P | [France] |
| Knaap et al., 2020 | J | | | P | Baltimore-Washington, USA |
| Milakis et al., 2018 | J | | N | N | [The Netherlands] |
| Nogués et al., 2020 | J | Expert survey | | N | [Spain] |
| Saghir and Sands, 2020 | J | | N | N | USA |

**Figure 3.** Summarized presentation of the effects on residential location forecast in the reviewed studies. S, P and N stand for "shared", "private" and "not specified", respectively. Source: Authors. Refs. [11,26–51].

Finally, two dominant anticipated impacts were also mentioned in the expert surveys, but these studies were also marked by a lack of agreement on these impacts. In their large survey targeting professional planners (N = 196), [42] showed that 49% of the respondents thought that the autonomous vehicle would encourage higher densities in the city center (26% disagreed) and at the same time 49% thought that it would result in increasing suburban sprawl (31% disagreed). Opinions were also divided on land-use effects in the study conducted by [43], which also reflected differing opinions on how the autonomous vehicle would mainly be deployed (e.g., private cars or not) and on the effects that would dominate (e.g., less need for off-street parking spaces or higher accessibility to distant locations). In a more policy-oriented survey, ref. [44] revealed a dominant skepticism about the positive impacts of AVs and pointed out the risk of urban sprawl.

### 3.2. Car Ownership

Among the studies that specifically examined the decisions to own (or relinquish) a car, the analysis of our corpus highlights four recurrent effects (Figure 4). The first and most frequent is a decrease in household motorization due to high-quality alternative modes, such as an automated mobility-on-demand service. Secondly, the studies also forecast a reduction in the motorization rate on the grounds that the development of private autonomous vehicles was likely to increase sharing practices within households (e.g., the same vehicle could drive one member of the household to their destination and then return home to drive another elsewhere). However, and thirdly, some studies concluded that there would be an increase in the motorization rate due to new users or new uses of the car, including people who are currently excluded from ownership due to age or physical disabilities, for example. Finally, and fourthly, some studies forecast a risk of lower motorization and possibly the social exclusion of low-income households, because of the combination of a ban on traditional vehicles in some areas and the high cost of autonomous vehicles.

| | Journal, Conference, Book or Other | Type of method | Decreased motorization due to alternative modes (including SAV) | Decreased motorization due to private sharing | Increased motorization due to new owners (new users → new owners) or new uses | Decreased due to social exclusion | Study area [when unspecified, countries of the authors to inform the geographical context] |
|---|---|---|---|---|---|---|---|
| Zmud and Sener, 2017 | C | Qualitative survey | N | P | N | | Austin, USA |
| Galish and Stark, 2021 | J | | No change (P) | P | P | | Berlin, Germany |
| Menon et al., 2019 | J | Quantitative survey | S | | | | USA |
| Kellett et al., 2019 | J | | S | | | | Adelaide, Australia |
| Kim et al., 2020a | J | | N | N | | | Georgia, USA |
| Liljamo et al., 2021 | J | | S | | | | Finland |
| Schoettle and Sivak, 2015 | O | Extrapolation | | P | | | USA |
| Bin-Nun and Binamira, 2020 | J | | S | | | | USA |
| Glus et al., 2017 | O | Modelling / extrapolation | S | | | | Cities in USA |
| Zhang et al., 2018 | J | Modelling | | P | | | Atlanta region, USA |
| Townsend, 2014 | O | Scenario analysis | S | | P | P | Cities in USA |
| Heinrichs, 2016 | B | | S | | | | [Germany] |
| Papa and Ferreira, 2018 | J | | S | | | P | [UK and Portugal] |
| Gruel and Stanford, 2016, 2017 | C, B | Scenario analysis supported by experts (workshop, survey) | S | | P | | [USA] |
| Pernestål Brenden et al., 2017 | O | | S | | P | | Sweden |
| Saujot et al., 2018 | O | | S | | | P | [France] |
| Enoch et al., 2020 | J | Expert survey | N | N | | | New Zealand, UK |
| Nogués et al., 2020 | J | | N | N | | | [Spain] |
| Saghir and Sands, 2020 | J | | N | N | | | [USA] |

**Figure 4.** Synthetic view of the effects on car ownership anticipated in the reviewed studies. S, P and N mean "shared", "private" and "not specified", respectively. SAV means Shared Autonomous Vehicles. Source: Authors. Refs. [9,26,27,30,41,42,44,46–50,52–59].

The first three effects were identified in some of the respondents' discourses in the qualitative survey reported by [26], whereas most of them did not expect to change the number of cars in their possession. Based on focus groups, the study conducted by [9] concluded that many people might not give up their private cars even if ride-sharing services based on automated vehicles became available. The authors also considered the effects of intra-household sharing practices and new owners, and highlighted the fact that the behaviors of the households will primarily depend on the terms under which

automated cars are implemented (especially pricing structures, legal regulations, and mobility services).

Four recent quantitative surveys have shown widely differing shares of respondents who reported favoring partial or full de-motorization: 14% in [52]; 26% in [53]; 37% in [27]; and 65% in [54]. Cultural specificities aside, these figures are nevertheless hard to compare because they considered contrasting scenarios (e.g., "Assume that all vehicles on the road are automated vehicles. Would you want/need to own a private automated vehicle, if an automated taxi would always be available within 5 min and the annual costs of automated taxis would be about 20% lower than the costs of a private automated vehicle" [54]), differing sample sizes and selection criteria (e.g., commuters working in the Central Business District for [52], and university personnel and members of the American Automobile Association for [53]). However, they provide valuable insights into the determinants of these choices (e.g., gender or socio-economic determinants).

The first two effects were also foreseen and quantified by extrapolation and modelling studies (Figure 4). By extrapolating data from the US National Household Transportation Survey (NHTS), ref. [30] found a significant decrease in car ownership (from 11% in rural areas to 37% in urban areas in terms of vehicles per household) when considering a future "where shared, on demand AVs are mostly suited for short trips in highly urbanized environments" [30]. It should be noted that this study has the particularity that it explicitly took account of the effect of changes in residential location on car ownership. Such changes were also modelled and quantified by [55] in the framework of a cost model approach, which focused in particular on the density of residential areas. The modelling approach was applied in conjunction with the simulation of modal choices for commuting in three American cities, concluding with contrasting results depending on the morphology of the cities (see Supplementary Materials). Adopting an agent-based modelling approach, ref. [56] concluded that the number of vehicles owned by the households could be reduced by 9.5% with a 30% increase in total distances travelled. Finally, ref. [57] focused on the second effect (sharing practices within the household) and arrived at a considerably higher estimate (a reduction of 43% was viewed as a high bound by the authors given the main assumption that vehicles could be shared among household members if their journeys do not overlap during the survey day).

The scenario-based studies largely forecast a decrease in motorization due to the deployment of shared autonomous services. However, only the study by [58] quantified the reduction. Moreover, three scenario-based studies from our corpus pointed up the risk of de-motorization due to social exclusion, because of the combined effect of a ban on traditional vehicles in some areas and the high cost of autonomous vehicles [41,46,48]. This would also have strong adverse impacts on access to mobility. In contrast, three studies foresaw an increase in motorization due to new uses or new users [46,49,50,58].

Finally, the three studies based on expert surveys all foresaw decreased motorization, even if the level of agreement varied among the interviewed experts [42,44,59]. Moreover, the cause of the decrease was not precisely identified. It is however the case that in these studies, the experts were also expected to describe the type of deployment they expected for autonomous vehicles (e.g., mainly shared or mainly private), and in some cases they were asked to take account of some other types of changes in their forecasts (e.g., mobility as a service, organizational, or infrastructural changes).

*3.3. Activity Patterns*

Our corpus reveals four recurrent impacts on activity patterns, in terms of their nature, duration or location. The effects may relate to any of these characteristics, but also to a more general change in individuals' daily programs. Activity patterns are greatly influenced by spatiotemporal factors, which can be revealed and studied by the discipline of time geography which provides an analytical framework for highlighting and understanding the complex spatiotemporal relationships between activities (see [60]). Unlike the previous

subsections, we have not separated the effects from the methods used, deeming that this provides greater clarity.

The first, and most obvious, anticipated effect consisted of new in-car activities (Figure 5). Via a focus group, ref. [10] identified a number of possible activities envisaged by commuters (e.g., reading newspapers, writing emails, and watching movies). In addition, ref. [61] analyzed a quantitative survey that identified several activities (such as private communication, online information search, passive entertainment, working, sleeping) and quantified the willingness to pay to perform them. Employing a fairly similar approach, ref. [31] underlined that reading, working, and studying were highly valued, unlike other activities (e.g., using the telephone, sleeping). The survey conducted by [62] reached very different conclusions: " [the] advantages of automated vehicles were predominantly identified for those activities already favored in today's conventional cars" [62]. The survey showed, in particular, only slight interest in working on board (only 13% of respondents valued this advantage, significantly less than leisure or social activities). In a large-scale empirical study, ref. [63] found a certain degree of interest in eating, sleeping, and working during in-car time, even if half of the respondents felt they were unlikely or very unlikely to change their activity patterns because of AVs. Furthermore, ref. [64] evaluated how individuals with long commuting times would use their in-car time in the automated vehicle era by studying the differences in activity patterns between them and other groups of people who were very similar but with shorter commutes. By using this analogy-based approach, they concluded that long-distance commuters had a latent demand for working, sleeping, and watching videos. The formal approaches implemented by [65,66] also considered changes in in-car activities but mainly provided insights into the impact of these changes in terms of the organization of the day's activity program (see below), since in-car activities can provide individuals with more flexibility. Finally, two expert-based approaches have also anticipated changes in in-car activities, mentioning in particular productive activities [67] or "resting, socializing, working, studying, eating, or merely contemplating the passing landscape" as the result of a speculative approach [48]. Although it only indirectly addresses this question (which explains why it is not reported in Figure 5), we can also mention the study by [68], who discussed the issue of in-car activities and the value of time in the automated vehicle era by analyzing current on-board activities.

The second effect consists of the reconfiguration of activity patterns resulting mainly from the first effect. This has been anticipated in seven of the studies in our corpus (Figure 5). Modelling approaches have provided a conceptual framework to explain and study such changes. Based on an approach relying on the Household Activity Pattern Problem (HAPP) model [65], or on a time-use model [66], or on an agent-based model [69], these studies examined the possible effects of performing activities in an autonomous vehicle (time reallocated to other activities, rescheduling). They also considered that some activities would no longer be performed (e.g., escorting children to school). They concluded that "This may lead to an increase, or decrease in travel time, depending on the traveler's preferences, schedule, and local accessibility" [66]. These effects were also identified and described in a focus group survey with commuters [10] and quantified in [63], who found respondents expected to "cultivate new hobbies or skills with the time [they] saved" [63]. Finally, such effects have also been mentioned in scenario analysis approaches, as a consequence of the time saved in the so-called optimistic scenario of [48], and as the consequence of the time freed up by autonomous vehicles due to their capacity to run errands for their owners [49].

| | Journal, Conference, Book or Other | Type of method | Change for new in-cars activities | Change in the organization of the day's program | Change due to easier access to new activities (especially for non drivers) | Changes due to change in the built environment | Study area [when unspecified, countries of the authors to inform the geographical context] |
|---|---|---|---|---|---|---|---|
| Pudāne, Rataj et al., 2018 | J | Qualitative survey | N | N | | | The Netherlands |
| Zandieh and Acheampong, 2021 | J | | | | S | | Manchester, UK |
| Cyganski et al., 2015 | C | Quantitative survey | P , S | | | | Germany |
| Dungs et al., 2016 | O | | N | | | | Germany, Japan |
| Carrese et al., 2019 | J | | N | | | | Roma, Italy |
| Kim et al., 2020b | J | | N | N | *Less change* | | Georgia, USA |
| Moore et al., 2020 | J | | | | P (work-place) | | Dallas, USA |
| Das et al., 2017 | J | Extrapolation | N | | N | | USA |
| Chen and Armington, 2016 | C | Modelling | P | P | | | [USA] |
| Pudāne, Molin et al., 2018 | J | | | N | | | [The Netherlands] |
| Vyas et al., 2019 | J | | | P | P | | Columbus, USA |
| Townsend, 2014 | O | Scenario analysis | | | | S | Cities in USA |
| Papa and Ferreira, 2018 | J | | P , S | P , S | P , S | S | [UK and Portugal] |
| Gruel and Stanford, 2016, 2017 | C, B | Scenario analysis supported by experts | | P | P | | [USA] |
| Fitt et al., 2019 | J | | | | S | | New Zealand |
| Nogués et al., 2020 | J | Expert survey | | | N | | [Spain] |
| Pettigrew et al., 2018 | J | | N | | N | | Australia |

**Figure 5.** Summarized presentation of the effects on activity patterns anticipated in the reviewed studies. S, P and N stand for "shared", "private" and "not specified", respectively. Source: Authors. Refs. [10,28,31,44,46,48–50,61–67,69–71].

The third effect anticipated by the reviewed studies is the opportunity to perform new activities, especially for people experiencing limited mobility. By analyzing in-depth interviews conducted with elderly people ($\geq$65 years), the analysis by [70] revealed in particular that "older adults perceive opportunities in AVs enhancing their physical activity, promoting social interaction". Adopting an approach by analogy, ref. [64] anticipated and quantified possible future changes in the activity patterns of a group of people with limited mobility by comparing their current practices with those of an otherwise similar cohort, in order to identify which currently sacrificed activities would become possible in the automated vehicle era. They identified shopping and socializing as the most affected activities. In their study, ref. [67] interviewed stakeholders (governments, academia, firms, etc.) across multiple countries and highlighted new opportunities in terms of access to jobs for people with mobility disadvantages (e.g., indigenous communities in Australia where the study was conducted). This kind of effect has also been depicted by scenario analysis approaches, in particular by [49,50] who considered new car users, by [48] in their so-called optimistic scenario and by [71] as part of a broader reflection focused on well-being. The latter considered that shared autonomous vehicles would increase accessibility as the high capital costs of these vehicles would be shared among many users. However, this conclusion has been qualified by the findings of [63], who showed that older people were more inclined to anticipate no change in their everyday activities in the autonomous vehicle era. Finally, some studies have considered the whole population rather than specific groups. In particular, ref. [63] showed intentions to change leisure activities, in terms of distance (e.g., travelling further for shopping or going to a restaurant) and frequency (e.g., more frequent travel to social/leisure activities). Moreover, ref. [28] highlighted that 30% of the respondents would be ready to change workplace despite a longer commute time in the autonomous vehicle era.

The fourth effect is mentioned in two imaginative scenario analysis approaches which considered shared automated vehicles. In this case, changes in the built environment generated changes in activity patterns. In their optimistic scenario, ref. [48] imagined that the space freed up by shared automated vehicles services could be used "to embellish the built environment or to respond to other societal needs beyond the transport sector", including "new recreational, building, and green areas, or converted into cycling or pedestrian infrastructures" [48]. Townsend [46] also envisaged a scenario with the gradual elimination of cars from the streets, freeing up space for parks, activities (sports grounds, etc.), urban farms, or alternatively, homes that would allow those living in them to perform new activities.

*3.4. Tourism*

Of all the types of impact, those on tourism have received by far the least attention in foresight studies (Figure 6). Although we found some perspective papers providing general insights, e.g., [72], these are outside the scope of this review. Our results are therefore based on a very limited number of studies.

| | Journal, Conference, Book or Other | Method | Changes of destination | Increase in the frequency of short duration trips | Change in modal choice | Study area [when unspecified, countries of the authors to inform the geographical context] |
|---|---|---|---|---|---|---|
| Gurumurthy and Kockelman, 2020 | J | Quantitative survey | | | N | USA |
| Kim et al., 2020b | J | Quantitative survey | N | N | | Georgia, USA |
| Lamondia et al., 2016 | C | Extrapolation | | | P | USA |
| Perrine et al., 2020 | J | Modelling | P | | P | USA |
| Cohen and Hopkins, 2019 | J | Scenario analysis | N | N | N | [United Kingdom] |

**Figure 6.** Summarized presentation of the effects on tourism anticipated in the reviewed studies. S, P and N stand for "shared", "private" and "not specified", respectively. Source: Authors. Refs. [19,63,73–75].

We have identified three frequently forecast effects. The first was a change in the destinations for leisure trips. This was anticipated first in the context of an imaginative approach by [19], who suggested that secondary cities or peripheral areas would gain in appeal as they became more accessible. This tendency was confirmed by the interest in more distant leisure travel identified in the survey analyzed by [63]. These two studies also mentioned the second effect (the reconfiguration of activity patterns), which consists of an increase in the frequency of short duration trips or vacations and in the number of destinations visited. The third effect was a change in transport mode for this type of trip. This was also foreseen by [19], who mentioned, in particular, the potential decline of intercity coach trips. It was also anticipated by [73], who studied modal changes for long-distance journeys, which account for a significant proportion of tourist activities. According to their hypotheses and extrapolation, approximately a third of journeys could be undertaken in an autonomous vehicle (with another third by plane and the rest by conventional vehicle) in a system dominated by private autonomous vehicles. This last effect was also investigated by means of a quantitative survey analyzed by [74], who found that nearly 50% of trips of between 50 and 500 miles (one-way) were expected to be performed in an autonomous vehicle. Using a long distance mode and destination model, ref. [75] forecast that the use of automated private cars would take market share away from air travel, resulting in an increase in the average distance travelled by private car but also a reduction in the average distance travelled by 6.7% in a year due to changes of destination as a result of less air travel. This study thus revealed both the third and the first effect.

## 4. Discussion

One of the main results of this review is the convergence of results across methods, especially for expected impacts on household location and car ownership and the influence of the way autonomous vehicles are deployed, often presented in terms of private versus shared vehicles. Choices related to household location and car ownership have numerous environmental and social impacts: levels of direct and indirect energy demand, greenhouse gas and pollutant emissions, urban land uptake, access to mobility and public space, to mention but a few. This provides additional confirmation of the size of the stakes involved in regulating the way autonomous vehicles are deployed, but also in terms of mitigating the negative effects of each type of deployment (e.g., limiting further growth in vehicle-kilometers in countries where it is already high). The second result comes from the nuances that appear in a second-order analysis, showing how important it is to consider a broad range of foresight methods when one wants to anticipate the effect of the deployment of autonomous vehicles. The multi-method analysis thus allows for dealing with uncertainty. Beyond this general finding, this study has provided a number of insights, relating on the one hand to the multi-method nature of the analysis, and on the other hand to the construction of knowledge about social change driven by autonomous vehicles.

### 4.1. The "Foresight Methods Wheel"

The production of the wheel below (Figure 7) allowed us to compare the different methods used, which are based on various foresight assumptions or processes but also have some principles in common, as apparent in the figure. We believe that this wheel helps to present a comprehensive and well-balanced picture of the effects that are frequently anticipated across the literature.

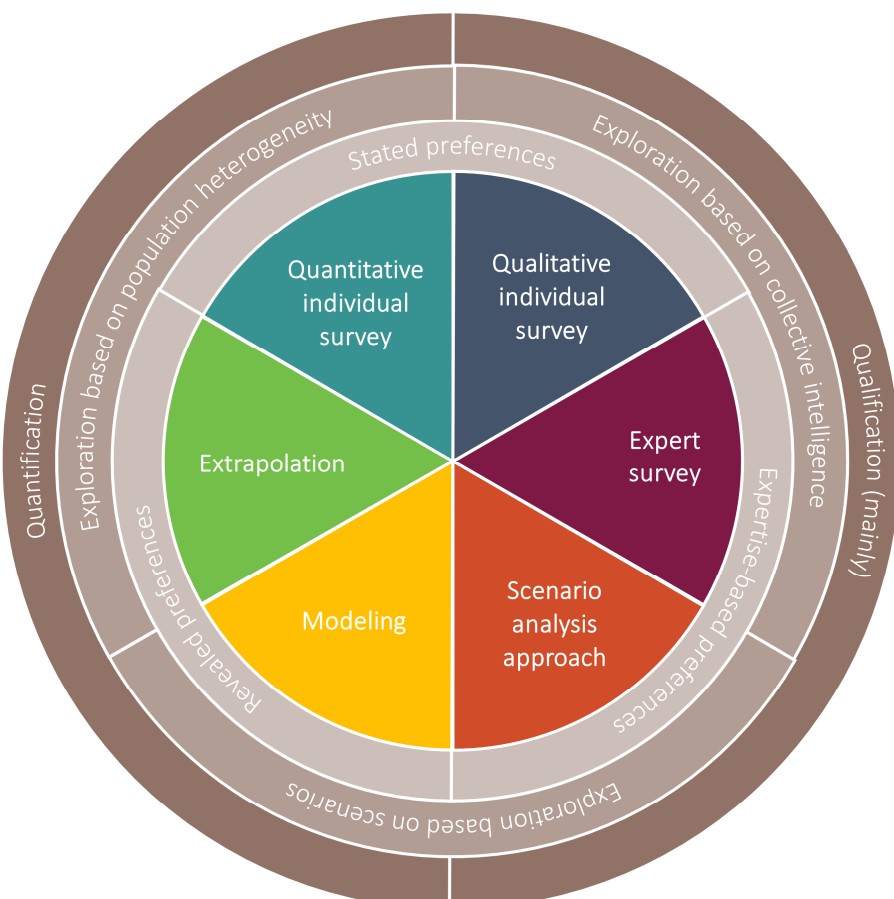

**Figure 7.** The foresight method wheel: a presentation of some the common and distinct principles underlying foresight methods. Source: Authors.

Basically, the methods in question are based on different assumptions. In stated preference surveys, the assumption is that individuals—who have varying degrees of knowledge about the issues—are able to anticipate their own future preferences and behaviors. In modelling and extrapolation approaches, the practitioners assume that future preferences and behaviors can be deduced from current practices. Extrapolations are usually based on the principle of analogy; for example, features of future mobility in the autonomous vehicle era are identified in today's world: e.g., non-driving mobility is considered on the basis of public transport or driverless vehicles [64], in order to decide on the types of trips for which an autonomous vehicle service would be efficient [30]. Modelling approaches consider that some of the relationships between variables that are observed today would remain valid in a future world, and that the new transport mode can be represented by combining the values of these variables (e.g., travel time, value of time, and speed). Finally, the last two families of methods assume that individuals who have a good knowledge of the current system and of its dynamics (i.e., experts) are able to estimate the effects of its disruption by the advent of a new mode.

These methods can also be characterized on the basis of other characteristics. In particular, three families of methods provide quantifications of the anticipated effects, whereas this is rare (or unaccompanied by any claim to representativeness) in the other three approaches (see Figure 7). The use of scenarios as a tool for exploring futures is particularly well-developed in the case of modelling and scenario analysis approaches. These two approaches also have the particularity of offering an explicit representation of the system at stake, either through formalization in the models or through the various representations which help to structure thought in the qualitative processes (e.g., $2 \times 2$ matrix techniques [40], a dynamic system model [49], a fuzzy cognitive map [76], or a conceptual framework [48]).

From a different standpoint, qualitative survey approaches (with experts or individuals drawn from society) provide an open process (generally involving open-ended questions and longer interview times), and in many of the studies in our sample, implement collective intelligence processes. Moreover, in our sample, when such processes involve experts, it is very common for them to be structured around a scenario development phase (so we have classified them as "hybrid").

Finally, extrapolation approaches and quantitative-stated preference surveys both base their expectations on population heterogeneity. Extrapolations aim to anticipate changes in the behavior of certain categories of the population by observing other categories. Examples we have identified include activity patterns in [64], car ownership in [57], and residential location in [30]. Quantitative-stated preference surveys aim to better understand behaviors and the factors that determine them based on the differences between groups of people, and may help to disseminate the lessons learned among the wider population.

The multi-method approach adopted in this literature review therefore makes it possible to evaluate and compare expectations that are ascertained on a variety of bases, making it possible to draw up a more complete and nuanced picture of the anticipated changes in lifestyles, which includes quantifications and identifies the main sources of uncertainty.

The methods each have their own areas of pertinence, strengths, and limitations. Scenario approaches, which provide processes for stimulating creativity while drawing on knowledge of the systems under study (often enhanced by contributions from groups), are the most exploratory as they generally consider a greater number of scenarios, dimensions, and effects. The anticipation of the risk of exclusion from vehicle ownership is an example of an effect that was only foreseen in these approaches [41,48]. Expert surveys also seem more conducive to the study of systems as a whole and varied effects. It is not uncommon for the autonomous vehicle to be only one of the disruptions addressed in these surveys (along with the advent of digital technology and vehicle sharing practices, for example). Qualitative household surveys are therefore particularly suitable for gaining a detailed understanding of the different dimensions that enter into the decision-making process. However, these approaches do not produce quantified results, which are particularly useful

for evaluating economic effects or greenhouse gas emissions, making a spatial or social disaggregation of effects or applying analyses to new areas, for example. On the other hand, we have observed that these methods often have the narrowest scope. Finally, we consider that all of these approaches can be valuable for informing public policies.

*4.2. How to Interpret and Deal with Divergences?*

Although, in general, our results indicate a high level of agreement between the conclusions of the various studies, we have also identified divergences between their predictions. In this case, it is questionable whether one approach can be authoritative. For example, modelling studies predict location changes due to a potential decrease in the value of travel time, but some surveys in which individuals are interviewed show many have little or no desire for such change [29]. This is also the case for the decision to de-motorize; most quantitative studies anticipate a decrease in car ownership due to the arrival of a new and competitive mode, whereas other surveys predict that households would still prefer to keep their private car(s) [9]. Such examples highlight the need for further studies that take into account the decrease in the value of time in autonomous vehicles (see [77]), and on research on the decision to de-motorize [78], which highlights specific determinants such as traffic or parking restrictions but also considers travel socialization.

In general, it is impossible to establish a hierarchy between the different types of methods, as they do not have the same ontological foundations or share the same hypotheses, as we have already seen. Modelling studies are often based on the hypothesis of rationality, whereas surveys of individuals or consultancy work by experts (including scenario analysis) place great emphasis on the role of representations in decision-making. It is widely recognized, for example, that individuals state they ascribe a higher value to the ability to travel and to perform activities made possible by car ownership than is apparent in their travel practices. Thus wishing to keep a car even if other modes offer more efficient solutions may also reflect a degree of rationality, even if this is not necessarily captured by the models.

However, within each family of methods, it can be argued that the results of some studies seem to be more informative about future changes. In stated preference surveys, it has been shown that survey design has a strong impact on the validity of the results. Some major aspects are: the level of knowledge provided to the respondents; the use of binary choice techniques rather than ranking; customizing (which consists of collecting information on a current day, or a current journey to present choices in relation to this reference); adaptive design (adapting choices to previous responses); and the use of threshold values (such as the value of time). Among the studies reviewed, it can be observed that as knowledge develops, survey designs become more sophisticated because they can draw on previous experience. The sample size of a survey is also decisive when aiming for representativeness. For modelling studies, the main hypotheses and their sensitivity are very important, one example being hypotheses about changes in the value of time. For extrapolation studies, one can mention the subtlety and sophistication of the starting hypothesis. For example, the central hypothesis in [57] to estimate the effect of autonomous vehicle deployment is probably cruder than those in [30,55]. However, they do not occur at the same point in the process of knowledge development, so the first study has more of a pioneering objective and is clear about the limits of its main hypothesis, indicating that the result is upper bound.

*4.3. Research Gaps*

Our study provides an overview of recent research on the effects of autonomous vehicle deployment in terms of lifestyle changes that shape travel behaviors. It appears that some aspects have not yet received much attention. This applies, first of all, to the changes in job choice which we had initially included in our analysis framework but which we finally included in the activity patterns framework because of the lack of studies. Yet, it is likely that a change in accessibility conditions will also have an effect on these

decisions. This is also the case for changes in tourism, as we had to base our results on a very limited number of studies. This lack of studies probably reflects the more indirect and uncertain nature of the effects of the arrival of a new transport mode on these decisions. However, it should be noted that empirical work has shown that these decisions are also widely linked to locational choices, daily activity patterns, and car ownership. Finally, it should be pointed out more generally that these interdependencies between the various components are also little addressed in our corpus and merit more attention in order to achieve a comprehensive analysis. For example, the effect of household relocation on car ownership has rarely been considered, particularly in quantitative terms (for an exception, see [30]). However, this effect is potentially important. Fewer studies have considered the links with activity patterns, and even fewer the link with tourism. We therefore believe that addressing these different aspects more as a bundle offers promising perspectives. By this we mean, for example, adopting a configurational approach towards lifestyles in order to better anticipate the effects of the deployment of the autonomous vehicle on our lives and travel behaviors.

*4.4. New Scenario Design*

We found that in most cases, the studies assume that one deployment mode for the autonomous vehicle will be dominated by either shared or privately-owned drivers. Although this is very useful for conceptualizing and describing the effects of the autonomous vehicle in early studies, we believe that the time has come to make projections more complex. We now need a better understanding of the competition and interaction between the different deployment modes, and, in particular, a more detailed analysis of the conditions in which they do or do not dominate (beyond basic preference mode choice surveys). It is indeed likely that a continuum of modes will coexist, providing distinct services under specific conditions (primarily according to the characteristics of land-use, but also according to the population). Moreover, the attractiveness of a mode may be greatly affected by how it is implemented. For example, on-demand autonomous mobility could be priced on a "per-ride" basis, as is the case today with taxis, or with a pass that provides unlimited travel within a given area.

Based on our findings, we also consider that there is a need to explore more scenarios that are compatible with environmentally and socially-oriented sustainable development goals. These should not be confined to the issue of electrification, which is not necessarily associated with autonomous vehicles. Some scenario analysis approaches have explored such perspectives, showing that the autonomous vehicle could help to eliminate the car from the city thereby assisting the creation of calm, multifunctional, public spaces, in particular: [41,46,48]. Adopting a lifestyle perspective is therefore a promising approach in order to consider all the benefits of such a future. However, this can only be possible if these objectives guide public action. Without regulation, private ownership, with its environmental and social costs, is most likely to become even more dominant. The issue is particularly important in developing countries where urbanization is expected to be much more rapid than in industrialized countries in the coming decades.

**Supplementary Materials:** The following supporting information can be downloaded at: https://www.mdpi.com/article/10.3390/futuretransp2030033/s1. Table S1: extended tables of the effects of autonomous vehicles on the four lifestyle components in the reviewed studies (related to Figures 3–6).

**Author Contributions:** Conceptualization, T.LG. and A.A.; methodology, T.LG.; formal analysis, T.L.G.; data curation, T.L.G.; writing—original draft preparation, T.L.G.; writing—review and editing, T.L.G. and A.A.; project administration, A.A.; funding acquisition, A.A. All authors have read and agreed to the published version of the manuscript.

**Funding:** This research was funded by DGITM (MTES). The funding from DGITM was provided under the subvention convention N° 17 SAGS-MTI 12-04.

**Data Availability Statement:** Data available on request.

**Conflicts of Interest:** The authors declare no conflict of interest.

## Appendix A

**Table A1.** Short description of the lifestyle components chosen.

| | Component | Brief Description |
|---|---|---|
| **Annual to pluriannual choices** | Choices about housing | This component covers the choice of a residential location (e.g., neighborhood, area type, region), type of housing (detached house, apartment), and its features (e.g., size, amenities). These choices are linked to demographic choices (e.g., parenthood) and cohabitation practices (e.g., living single or in a couple, shared rental accommodation), which reflect the structure of family models, even if they are not considered here. |
| | Choices about work and income | Choices about work and income cover the choice between free time and earnings, generally corresponding to working hours, distribution of labor (within the household, over the life cycle), constraints or flexibility linked with workplace(s) (including homeworking), and other sources of income (e.g., interest on savings). |
| | Choices about vehicle ownership | This component relates to the vehicles households own and share (especially cars), their type and number, as well as the forms of access or renewal (purchase, rental or borrowing). These items provide access to services and to activities. |
| **Daily to annual choices** | Day-to-day activities | This component includes the choice of activities and their characteristics. The activities in question may be imposed on individuals (e.g., household tasks) or voluntary (e.g., leisure, visits to friends). The characteristics of the activities relate essentially to their frequency and duration, location (e.g., at home, in the neighborhood), and the physical and virtual components of activities and interactions. The virtual component is linked with the use of telecommunications devices which, since the first telephone conversations, have profoundly altered the structure of activities and real mobility needs, without, however, replacing them. |
| | Vacations and holidays | Choice regarding vacations and holidays refers to tourism and other leisure activities, as well as visits to family or friends, which fall outside the category of day-to-day activities. These choices are characterized by their frequency, duration and the holiday destination, the associated expenditure, and by the activities undertaken (e.g., cultural tours, sports). |
| **Travel behaviors** | Mobility practices and relationship with space | Day-to-day and exceptional activities are made possible by mobility practices which do not appear on the same level in the proposed representation, but are an integral part of the studied decision-making system. Mobility is an activity that is not pursued for its own sake (economists talk of transport as "derived demand"), but the associated opportunities, constraints and choices are incorporated within activity choices (see for example certain analyses in the field of transport economics [79]). However, some of the choices—for example transport mode—may be specific to particular mobility practices, even though they may be constrained.<br><br>All the places with which individuals interact in their activities constitute their living space, which generally varies from the scale of the neighborhood or village to that of the region for day-to-day activities, and goes beyond these for exceptional activities (e.g., long-distance travel, tourism). |

Source: Authors.

## Appendix B. Supplementary Information Related to the "Identification Phase" Implemented to Constitute the Corpus

Within the identification phase (Figure 2), we refined the research by the following subject areas (Scopus) or categories (Web of sciences):

1. Subject areas in Scopus: Social Sciences; Environmental Science; Decision Sciences; Energy, Economics and Finance; Psychology; Arts and Humanities; Multidisciplinary; and Undefined.
2. Categories in Web of Science: web of science categories: social sciences interdisciplinary; behavioral sciences; planning development; transportation; environmental sciences; energy fuels; psychology; social issues; green sustainable science technology; psychology multidisciplinary; urban studies; engineering environmental; ecology; multidisciplinary sciences; geography physical; statistics probability; philosophy; environmental studies; information science library science; geography; and economics.

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
