# Peer review of "Anticipating Changes in Lifestyles That Shape Travel Behavior in an Autonomous Vehicle Era—A Method-Oriented Systematic Literature Review"

_futuretransp, doi:10.3390/futuretransp2030033_

Round 1
Reviewer 1 Report
Thank you for the opportunity to read your paper. Although It is a well-written paper...but I don't think you add much to the literature on what is known on the topic. Your results are varied and inconclusive.... as is reflective of the studies done on this topic... You add lifestyle changes and the forecasting methods (which is just essentially the methods used in these studies) and there is no consensus (I didn't expect it either). So it is hard to see what new knowledge you are adding to this field.

Author Response
Thanks for your comments. One of the main results of this review is the convergence of results across methods, especially for expected impacts on household location and car ownership and the influence of the way autonomous vehicles are deployed. The second result comes from the nuances that appear in a second-order analysis, showing how important it is to consider a broad range of foresight methods when one wants to anticipate the effect of the deployment of autonomous vehicles. The multi-method analysis thus allows for dealing with uncertainty.
Reviewer 2 Report
Relevant topic, very well written, new insights generated
Just very few minor editorial comments:
- Abstract (lines 10-12): Sentence grammar/incomplete
- Sometimes reference missing: line 138, 209, 321
- Few times et al. dot is missing: line 222,225
Author Response
Relevant topic, very well written, new insights generated
Thank you very much for your positive comment. We really appreciate.
Just very few minor editorial comments:
- Abstract (lines 10-12): Sentence grammar/incomplete
It has been corrected.
- Sometimes reference missing: line 138, 209, 321
Thank you. References were added.
- Few times et al. dot is missing: line 222,225
Thank you. It was corrected.
Reviewer 3 Report
The manuscript conducted a method-oriented systemic literature review for anticipating changes in lifestyles that shape travel behavior in the autonomous vehicle era. The topic of the manuscript is suitable for the journal. The overall manuscript is well written. The reviewer has following suggestions.
- It would be excellent if the authors could analyze whether there is a similarity if studies are conducted in the same country (or continent). Did the authors find any similarities and differences across the countries?
- There are some grammatical errors in the manuscript. For example, Line 10: Correct “Its discusses”.
- For 2.2. constitution of the corpus: it will be helpful if a flowchart is included to explain the framework for the better and clear understanding.
- Figure 2: Venn diagram would be more suitable. The size of each circle can represent the number of papers by method and show the hybrid approaches.
- Figure 4: “SAV” was not defined in the manuscript. It should be added in the table (as a note).
- “AVs” was not defined in the manuscript.
Author Response
- It would be excellent if the authors could analyze whether there is a similarity if studies are conducted in the same country (or continent). Did the authors find any similarities and differences across the countries?
Thanks for pointing this out. By specifying the countries to which the studies refer in Fig. 3-6, we intended to provide information on the context of the study. However, our analysis showed that we can’t find any significant trend related to this parameter in our corpus (either similarities or differences). We thus add the following sentences to the introductory paragraph of the result section.
“As this information is also reported below, it is worth noting that we also analyzed potential similarities and differences across the countries in which were conducted the studies. However, we didn’t find any significant trend in our corpus, the type of deployment appeared to be much more explanatory. We can only highlight a slight over-representation of the private ownership assumption in US studies.”
- There are some grammatical errors in the manuscript. For example, Line 10: Correct “Its discusses”.
Thank you, that has been done.
- For 2.2. constitution of the corpus: it will be helpful if a flowchart is included to explain the framework for the better and clear understanding.
Thanks for this suggestion. We have added such a flowchart in the main text.
- Figure 2: Venn diagram would be more suitable. The size of each circle can represent the number of papers by method and show the hybrid approaches.
Thanks for this suggestion. We produced a Venn diagram as recommended and combined it with the flowchart. We hope it is clearer for readers than the previous chart (also pasted below for comparison).
- Figure 4: “SAV” was not defined in the manuscript. It should be added in the table (as a note).
This has been done.
- “AVs” was not defined in the manuscript.
This has been done.
Round 2
Reviewer 1 Report
Thank you for the revised paper, but in my opinion it still does not highlight your unique contribution to the topic. You mention it in your response and yet you did not think it was necessary to highlight it in your paper.
Author Response
Thanks for your encouragement and suggestions. We agree that the main originalities and results of the paper and deserve a better highlight. As you can see in the revised version:
- We rewrote the first sentence of the abstract to highlight the novelty of the lifestyle perspective;
- We added a sentence in the introduction to highlight the focus on the multi-method approach we adopted in this study;
- We highlighted the main results in the abstract and at the beginning of the conclusion section.
Reviewer 3 Report
Thanks for taking the reviewer's suggestions and addressed the manuscript based on them.
There are a few more comments:
Lines 202: do not use a contraction like "didn't" in academic writing.
Figure 2: In the Venn diagram, the circles' size should reflect the number of studies. For instance, 'modelling' should be 3.5 times larger than that of 'extrapolation' and 'qualitative survey'.
Author Response
Thanks for your comments. We corrected the line 202. Regarding the Venn diagram, we thank you for your suggestion and updated the diagram. As it was more elegant, we considered the area of the discs to be proportional to the number of studies (including the “hybrid” ones) rather than the radius.
Round 3
Reviewer 1 Report
Thank you for revising your manuscript, but it is still not a substantive revision.